# Skin Microbiome and Radiation-Induced Skin Injury: Unraveling the Relationship, Mechanisms, and Therapeutic Implications

**DOI:** 10.3390/ijms26115022

**Published:** 2025-05-23

**Authors:** Aleksandra Wiktoria Bratborska, Paweł Głuszak, Maria Joks, Joanna Kaźmierska, Jakub Pazdrowski, Adriana Polańska, Shalini Jain, Hariom Yadav, Michal M. Masternak, Aleksandra Dańczak-Pazdrowska

**Affiliations:** 1Department of Dermatology and Venereology, Poznan University of Medical Sciences, 60-355 Poznan, Poland; mariajoks@wp.pl (M.J.); apolanska@ump.edu.pl (A.P.); 2Doctoral School, Poznan University of Medical Sciences, 61-701 Poznan, Poland; paw.gluszak@gmail.com; 3Department of Dermatology, Poznan University of Medical Sciences, 60-355 Poznan, Poland; aleksandra.danczak-pazdrowska@ump.edu.pl; 4Department of Radiotherapy, Poznan University of Medical Sciences, 61-866 Poznan, Poland; joanna.kazmierska@wco.pl; 5Department of Head and Neck Surgery, Poznan University of Medical Sciences, 61-866 Poznan, Poland; jakub.pazdrowski@wco.pl (J.P.); michal.masternak@ucf.edu (M.M.M.); 6USF Center for Microbiome Research, Microbiomes Institute, University of South Florida Morsani College of Medicine, Tampa, FL 33612, USA; jains10@usf.edu (S.J.); hyadav@usf.edu (H.Y.); 7Burnett School of Biomedical Sciences, College of Medicine, University of Central Florida, Orlando, FL 32827, USA

**Keywords:** skin, cancer, microbiome, radiation, injury

## Abstract

Radiotherapy (RT) is a treatment method commonly used in oncology. A vast majority of patients undergoing RT suffer from radiation-induced skin injury (RISI), which results from complex biochemical reactions in the irradiated skin. Current strategies for preventing and managing RISI are insufficient for achieving full skin regeneration. Multiple studies have shown that alterations in the skin microbiome correlate with the development and severity of RISI. These studies suggest that dysbiosis is a crucial factor in promoting radiation-associated dermatitis. Targeting the skin microbiota presents a potential therapeutic approach that could significantly improve the quality of life for patients undergoing RT. This review aims to present current findings on the interplay between the skin microbiome and radiation-induced skin damage as well as to discuss potential therapeutic strategies for preventing and mitigating this condition.

## 1. Introduction

Radiotherapy (RT) is a valuable treatment in cancer management, leveraging ionizing radiation to selectively target and destroy malignant cells. More than half of all cancer patients receive RT as a part of their treatment regimen [1]. The therapeutic efficacy of RT contributes to approximately 40% of tumor control in multimodal treatment approaches. The specific frequency, duration, and combination with other treatments depend on multiple factors, including the type and stage of the cancer, the patient’s overall health, and the treatment goals, as well as radiation dosages and duration. However, the primary problem is the damage it causes to the normal tissues surrounding the malignant tumor [2]. Observed side effects of RT include chromosomal aberrations, secondary cancers, infertility, and damage to internal organs and skin [3,4,5].

One of the prevalent adverse effects of RT is radiation-induced skin injury (RISI), also described as radiodermatitis or radiation dermatitis, which at some levels affects up to 95% of patients undergoing RT. Acute RISI (aRISI) manifests within the first 90 days of radiation treatment and presents as erythema, pigmentation changes, edema, and dry or moist desquamation. It is noteworthy that, in severe cases, aRISI may necessitate temporary or permanent cessation of RT, jeopardizing the success of the treatment. Chronic RISI, on the other hand, can appear months to years post treatment and includes symptoms such as skin hypersensitivity, dyspigmentation, xerosis, telangiectasia, alopecia, fibrosis, ulcers, and radiation-induced morphea (RIM) [6,7]. Although chronic RISI does not interfere directly with the effectiveness of RT, it significantly impacts the patient’s quality of life. The risk and severity of RISI are influenced by several factors, including higher radiation doses per fraction, greater cumulative doses, concurrent chemotherapy or immunotherapy, and treatment in anatomically sensitive regions in areas with thin skin or skin folds, such as the head, neck, breast, and axilla [8,9].

The skin microbiome, a vast and diverse community, has been suggested to play a key role in the development and progression of RISI [10]. The aim of this review is to present the current findings on the interaction between the skin microbiome and radiation-induced skin damage and to discuss potential therapeutic strategies for its prevention and management.

## 2. Mechanisms Underlying Radiation-Induced Skin Injury

The RT-induced damage occurs at the molecular level, leading to extensive DNA damage and generation of reactive oxygen species (ROS), which disrupt critical cellular metabolic processes and induce a complex cascade of signaling pathways [11]. Within the skin, these effects initiate a cascade of inflammatory responses, including the activation of nuclear factor kappa B (NF-κB) and the release of chemokines, adhesion molecules, and pro-inflammatory cytokines, including eotaxin, intercellular adhesion molecule 1 (ICAM-1), interleukin (IL)-1, IL-3, IL-5, IL-6, IL-8, and tumor necrosis factor-alpha (TNF-α). Together, they contribute to endothelial cell damage, increased vascular permeability, and immune cell recruitment, ultimately leading to local inflammation and skin breakdown [12]. Monocyte migration to the irradiated skin sites results in their differentiation into macrophages, which secrete platelet-derived growth factor (PDGF) and transforming growth factor-beta (TGF-β). These factors, in turn, promote the migration of fibroblasts and the activation of pro-fibrotic pathways [13].

The ionizing radiation damages the DNA, which results in malfunction, necrosis, apoptosis, and cellular senescence as well as local inflammation [14,15,16]. Additionally, it contributes to the formation of ROS and Reactive Nitrogen Species (RNS) through the radiolysis of water and by damaging the respiratory chain of the mitochondria [17,18]. The disturbed balance between oxidants and antioxidants leads to oxidative stress, directly damaging cells and creating a pro-inflammatory environment conducive to further biochemical reactions. Simultaneously, the damage-associated molecular pattern (DAMP) molecules released from destroyed cells recruit immune cells, with neutrophils being the first to be attracted to the site of inflammation. As they transfer into the pro-inflammatory phenotype, they release numerous interleukins, cytokines, and chemokines, with TNF-α and TGF-β being the most significant, as they constitute the main trigger factors for the development of fibrosis [18,19]. The released chemokines exacerbate local inflammation, leading to the further recruitment of immune cells.

The role of macrophages during the inflammatory process following RISI is ambiguous. Macrophages activated classically (M1) through DAMP molecules, infections, or interferon, release proinflammatory cytokines such as IL-1β, IL-12, and TNF-α. Initially, M1 macrophages promote fibrinolysis by excreting enzymes degrading collagen. However, prolonged activation of M1 macrophages has been associated with a pro-fibrotic effect, as it promotes the transition from fibroblasts and pericytes to myofibroblasts. As a result, an excessive amount of extracellular matrix (ECM) is produced, leading to the development of fibrosis [14,18,20]. Conversely, macrophages activated alternatively (M2) through IL-4 and IL-13 are proven to have anti-inflammatory and pro-remodeling effects [18,20]. Their presence is more pronounced in the later stages of inflammation, and the overall abundance of M2 macrophages is lower compared with M1 macrophages. Nevertheless, significantly higher levels of M2 macrophages have been observed in fibrotic diseases [18]. Chen et al. investigated the impact of senescent cells in re-epithelialization in RISI. According to his research, senescent cells excrete IL-33, which triggers the macrophage polarization into the M2 subtype [15]. Further research needs to be performed; however, this might explain the disrupted balance between M2 and M1 macrophages in the course of radiotherapy. The production of TGF-β, connective tissue growth factor (CTGF), and IL-13 makes them pivotal agents in fibrogenesis. A higher abundance of M2 triggers excessive fibrotic tissue accumulation and suppresses proper remodeling [14,16,18,20]. The imbalance between M2 and M1 is believed to play a critical role in the transition from acute dermatitis to chronic fibrosis [14].

Additionally, fibrosis development depends on the activation and differentiation of T cells [14,21]. Among numerous cytokines released by Th1 cells, the crucial one is interferon (IFN)-γ, being responsible for blocking TGF-β-induced myofibroblast transformation [22]. As a result, the ECM production is suppressed [18]. According to the research of Linard et al., Th1 cell deficiency is consecutively associated with a lower level of IFN-γ, a higher level of TGF-β, and a higher susceptibility to radiation-induced fibrosis (RIF) [23]. While Th1 cells have a suppressive effect on fibrogenesis, Th2, Th9, Th17, and Th22 cells are positively associated with RIF. The cytokines released from activated Th17 and Th2 cells, i.e., IL-17 and IL-13, have a pro-fibrotic effect by enhancing the TGF-β-initiated ECM production and by contributing to the development of a pro-inflammatory environment that promotes fibrogenesis [14,19,21,22].

Due to their high sensitivity to RT, information about the influence of B cells in RIF is scarce [18]. However, their role is mainly focused on maintaining the inflammatory environment by modulating other immune cells and fibroblasts [14,24].

Furthermore, skin damage induced by RT includes direct destruction of the skin layers. A prospective study conducted by Pazdrowski et al. revealed statistically significant differences in transepidermal water loss (TEWL), an indicator of the compromised epidermal barrier, in irradiated skin across various time points [25]. Furthermore, in patients who had previously undergone RT for head and neck cancer, TEWL was significantly elevated in irradiated regions compared with non-irradiated areas. Notably, the median time since RT was 6 years, and increased TEWL was observed irrespective of the presence of clinical manifestation of cRISI [26].

The intact epidermal lipid barrier plays a crucial role in inhibiting the overgrowth of pathological microbiota due to the antibacterial properties of skin fatty acids [27]. Additionally, for many skin commensals, skin lipids serve as an essential nutrient source [28]. Therefore, damage to the skin barrier induced by RT is a plausible factor contributing to alterations in the skin microbiome in cancer patients. Figure 1 illustrates changes in skin cells and cell signaling following RT.

## 3. Skin Microbiome

Skin, the largest organ of the human body, serves as a protective barrier against environmental factors. It is estimated to harbor thousands to millions of microbial cells per square centimeter, depending on the specific region. This diverse microbial population includes bacteria, viruses, fungi, and microeukaryotes (e.g., mites), which co-exist in symbiotic relationships with the host. Numerous internal and external factors influence the distribution and abundance of these microbial communities, including age; sex; hormone levels; stress; climate; exposure to ultraviolet (UV) radiation, pollution, or chemicals; as well as hygienic and cosmetic practices [29,30,31]. Additionally, the local composition of glands and hair follicles affects bacterial colonization in different body regions. Sebaceous areas such as the face and back are enriched with lipophilic *Cutibacterium* species. Moist areas, including the axillary vault, interdigital spaces, and inguinal crease, favor the growth of *Corynebacterium* and *Staphylococci* species. In contrast, dry areas like the inner forearms are more commonly colonized by *Proteobacteria* and *Flavobacteriales* [32]. Among fungi, *Malassezia* is the most prevalent genus, accounting for 80% of the skin fungal flora [33], and is particularly dominant in sebum-rich areas such as the face, trunk, and scalp [34]. *Demodex* mites, a type of microeukaryote, inhabit pilosebaceous follicles, predominantly on the face [35]. Viruses remain the least-studied component of the skin microbiome, with the majority being bacteriophages belonging to families such as *Caudovirales*, *Siphoviridae*, and *Myoviridae* [36].

The presence of the commensal microbiota contributes to the upregulation of genes associated with immune and inflammatory responses as well as keratinocyte differentiation. Skin colonization by microorganisms stimulates the production of proinflammatory cytokines such as IL-1α and IL-1β by immune cells. Furthermore, the commensal microbiota modulates epidermal proliferation and differentiation by influencing the gene expression of structural proteins such as filaggrin, repetin, and psoriasin [37].

Importantly, the skin microbiota plays an essential role in maintaining the skin’s barrier function. For example, *Staphylococcus epidermidis* produces sphingomyelinase, an enzyme that facilitates the host’s synthesis of ceramides, waxy lipid molecules that prevent dehydration [38]. In addition, microbes are also responsible for secreting agents that activate aryl hydrocarbon receptors (AHRs) in keratinocytes, supporting epidermal differentiation and skin integrity [39].

Skin also maintains microbial balance through antimicrobial peptides (AMPs) and enzymes that regulate the skin’s pH and moisture levels. Recent studies indicate that the skin also functions as a neuro–immuno–endocrine organ, integrating environmental signals to regulate both local and systemic homeostasis, including preserving skin integrity and adaptation to environmental changes [40]. Defensins, including human neutrophil peptides (HNPs), are a class of AMPs secreted by both keratinocytes and immune cells during inflammation. These peptides exhibit broad-spectrum antimicrobial activity, directly targeting pathogens and preventing their colonization [41].

Furthermore, human skin is an active immune organ populated by various immune cells, including Langerhans cells, dermal dendritic cells, macrophages, mast cells, and different subtypes of T cells and B lymphocytes [42]. Immune cells within the skin interact dynamically with the skin microbiota, and this mutual relationship is crucial for maintaining skin homeostasis. *Staphylococcus epidermidis* has been shown to activate gamma delta (GD) T cells and induce the expression of antimicrobial perforin-2 (P-2) [43]. In murine models, early life colonization of skin with *Staphylococcus epidermidis* promotes the activation of regulatory T (Treg) cells in the neonatal skin, thereby establishing immune tolerance to commensal microbes [44]. Interestingly, neonatal colonization with *Staphylococcus aureus* but not with *Staphylococcus epidermidis* upregulates IL-1β expression and increases the ratio of T helper 17 (Th17) cells to Tregs, suggesting a more inflammatory immune response [45]. Furthermore, commensal colonization with *Staphylococcus epidermidis*, *Staphylococcus xylosus*, *Staphylococcus aureus*, *Corynebacterium pseudodiphtheriticum*, and *Cutibacterium acnes* leads to an accumulation of IL-17A- and IFN-γ-expressing T cells in the skin, which in turn upregulates the expression of the antimicrobial alarmins S100A8 and S100A6 [46]. Therefore, colonization with commensal species is a crucial element of effective protection against invasive microbes. Keratinocyte expression of major histocompatibility complex class II (MHCII) is another factor contributing to homeostatic immunity to commensal colonization, primarily through the accumulation of Th1 cells in the skin [47]. Importantly, T cells induced by *Staphylococcus epidermidis* have been demonstrated to accelerate wound healing in mice [48]. Moreover, *Staphylococcus epidermidis* has been shown to activate T cells and upregulate perforin-2 expression, therefore increasing the ability of skin cells to kill intracellular *Staphylococcus aureus* in human skin ex vivo [43]. In addition, this species also produces lipoteichoic acid (LTA), which activates TLR2 and exerts anti-inflammatory effects on keratinocytes, thus supporting barrier function and immune homeostasis [49]. Interestingly, *Cutibacterium acnes* regulates immune tolerance through the production of short-chain free fatty acids (SCFAs), which inhibit the activity of histone deacetylase (HDAC) 8 and 9, and therefore downregulate the expression of pro-inflammatory IL-6 and IL-8 [50]. Moreover, commensal bacteria produce siderophores, which are small metal-chelating agents that facilitate iron acquisition. By competing with pathogenic bacteria for this essential nutrient, siderophores play a crucial role in inhibiting the growth of harmful microbes [51].

Importantly, rare environmental pathogens such as *Corynespora cassiicola*, although primarily classified as a plant pathogen, can exhibit opportunistic pathogenic behavior under specific environmental conditions [52]. *Candida auris* can colonize the skin and survive in the hospital environment, leading to subsequent transmission between individuals and to infections, especially in patients with indwelling devices [53]. Interestingly, certain fungi species, such as *Fonsecaea monophora*, can colonize the skin by producing melanin, which inhibits the immune response and leads to pathogen overgrowth [54].

This evidence altogether indicates that alterations in the skin microbiome, accompanied by a disrupted skin barrier, increase the susceptibility to multiple skin diseases. On the other hand, the presence of inflammation in different skin disorders significantly contributes to dysbiosis [55].

## 4. Skin Microbiota in RISI

Studies have shown that RT alters the skin microbial barrier by significantly reducing its abundance and diversity. It is noteworthy that the composition of the skin microbiome before the beginning of RT significantly impacts the occurrence and severity of RISI, providing a possible prediction for the disease outcome. However, the results of studies conducted so far are inconclusive. Research by Huang et al. on aRISI rat models revealed a significant predominance of Firmicutes, especially *Streptococcus*, *Staphylococcus*, *Acetivibrio ethanolgignens*, *Peptostreptococcus*, and *Anaerofilum* in rats that developed aRISI after RT compared with the control group, with no previous contact with RT. Researchers additionally analyzed patient data from BioProject 665,254 and observed an overall significant reduction in bacterial diversity following RT as well as a greater abundance of *Klebsiella*, *Pseudomonas*, and *Staphylococcus* in patients with RISI compared with healthy subjects. Interestingly, the analysis revealed a significant predominance of Proteobacteria and a low abundance of Firmicutes after RT in the group of patients who developed chronic ulcers [56].

Another study explored the cutaneous microbiota of 78 patients with RISI, both acute and chronic. Compared with the control group with no RT history, RISI patients exhibited a predominance of Firmicutes and Proteobacteria. RISI was associated with a predominance of *Klebsiella*, *Staphylococcus*, or *Pseudomonas*, while the skin of healthy subjects was mainly inhabited by *Klebsiella*, *Cutibacterium*, *Corynebacterium*, *Bacillus*, and *Paracoccus.* In addition, a longer duration of RISI was negatively correlated with the diversity of cutaneous bacteria. Slower healing of RISI was associated with greater amounts of *Pseudomonas*, *Staphylococcus*, and *Stenotrophomonas*. Consistent with the previous study, chronic ulcers were linked to the predominance of Proteobacteria and a low abundance of Firmicutes. The skin microbiota of these patients consisted mainly of *Klebsiella* or *Pseudomonas*, *Cutibacterium*, and *Stenotrophomonas*. The coexistence of *Pseudomonas*, *Staphylococcus*, and *Stenotrophomonas* was strongly correlated with the development of chronic ulcers [10].

Another study exploring skin microbiota in RISI detected a significantly higher abundance of *Ralstonia*, *Truepera*, and *Methyloversatilis* genera and a lower abundance of *Staphylococcus* and *Corynebacterium* genera in patients with no/mild aRISI (RTOG 0/1) compared with patients with severe aRISI (RTOG 2 or higher), both before and after RT [57]. On the other hand, research by Hülpüsch et al. revealed an association between a low number of commensal skin bacteria, i.e., *Staphylococcus epidermidis*, *Staphylococcus hominis*, and *Cutibacterium acnes*, at the beginning of the treatment and the development of severe aRISI. Additionally, a non-species-specific overgrowth of skin bacteria has been proven to occur right before the onset of RISI symptoms [58]. Similarly, another study assessed the composition of cutaneous *Staphylococcus* species before RT and linked the low abundance of *Staphylococcus hominis* and *Staphylococcus aureus* to the development of severe aRISI [59]. In addition, research by Kost et al. explored the impact of nasal colonization with *Staphylococcus aureus* before RT on the development of aRISI in patients with breast or head and neck cancer. The baseline colonization with *Staphylococcus aureus* in the nares was higher in patients who developed grade 2 or higher aRISI compared with those with grade 1. Interestingly, after RT, the *Staphylococcus aureus* colonization was higher in the nares, irradiated skin region, and contralateral skin in patients with grade 2 compared with patients with grade 1 aRISI [60].

Ulceration is one of the most severe clinical manifestations of RISI. Acute ulcers are less frequent and develop on the base of wet desquamation. Conversely, chronic ulcers typically occur in the later stages of the disease [61]. Patient-related risk factors for ulcer development include concomitant diseases and a particular composition of the skin microbiota, which, as mentioned above, exhibits several differences when compared with RISI patients without chronic ulcers [10,56]. Although the ulceration is a clinical manifestation of RISI, assumptions about its microbiome should not be extrapolated solely from data regarding typical bacteria in RISI. Table 1 summarizes studies on microbiota in RISI.

It is essential to highlight the bidirectional influence of RISI and the skin microbiome. On the one hand, RT induces a cascade of events that cause alterations in immune cells and damage to the skin barrier, subsequently leading to dysbiosis. On the other hand, changes in the proportion of different microorganism species residing on the skin have been linked to the development of various types of dermatosis, such as atopic dermatitis (AD) and seborrheic dermatitis (SD), among others, and therefore could potentially aggravate RISI. Apart from significantly reducing the diversity of skin microorganisms, the cause-and-effect sequence between RT and the skin microbiome needs further investigation.

Overall, the findings suggest a significant impact of RT on creating a potentially favorable environment for the excessive proliferation of pathogens, and as a result, for an exacerbation of the inflammatory process and severe skin injuries. First of all, a few studies showed that the predominance of bacterial species from the Firmicutes and/or Proteobacteria phylum was associated with prolonged healing of aRISI. The most frequently detected genera of cutaneous microbiota in patients with aRISI were *Staphylococcus*, *Klebsiella*, and *Pseudomonas*. On the other hand, research linked the low abundance of *Staphylococcus* species before RT, specifically *Staphylococcus epidermidis*, *Staphylococcus hominis*, as well as *Staphylococcus aureus*, to either the development of aRISI or a severe course of aRISI, suggesting that the cutaneous microbiota composition before RT might be one of the predictors of the RISI course. The major limitation of certain studies is the absence of specification of the exact Staphylococcus species that are overgrown in RISI patients. This information could provide a better understanding of the microbiota’s characteristics both before and after radiotherapy, as well as its influence on the clinical outcomes. Further research focusing on skin microbiota is needed to help identify these associations. It is noteworthy that the results were unequivocal regarding the predominance of Proteobacteria and the low abundance of Firmicutes in patients who developed chronic ulcers. In addition, it is important to note that the aforementioned studies exhibited a lack of procedural standardization. Variations in sampling locations, temporal factors, and microbiological analysis methodologies may compromise the validity of the experimental outcomes. Furthermore, the limitations of certain studies can be attributed to small sample sizes. Notably, the research included both animal models and human studies, which requires cautious interpretation when extrapolating the results to clinical contexts.

## 5. Management of RISI by Supporting the Skin Microbiome

### 5.1. Skin Care Products

Implementing preventative actions might alleviate severe cases of aRISI and improve patients’ condition. Proper skin care is well established and regarded as essential in the prevention and treatment of RISI. The skin should be washed with gentle cleansing products that do not disrupt the hydrolipid barrier, such as synthetic detergents (syndets), while concurrently using emollients to maintain skin moisture and UV protection. It is noteworthy that washing irradiated skin solely with water during RT is associated with increased severity of RISI as well as a higher frequency of moist desquamation and itching compared with washing with water and mild soap [62].

Emollients are fundamental in the treatment of AD, which, as mentioned before, shares several pathophysiological similarities with RISI [63,64]. Emollients are composed of a mixture of lipids, typically in a 3:1:1:1 ratio of cholesterol, ceramides, essential free fatty acids, and non-essential free fatty acids. Additionally, they may contain other lipids, such as mevalonic acid, which has been demonstrated to accelerate the restoration of the hydrolipid barrier. Emollients in AD have been shown to reduce TEWL and restore the hydrolipid barrier, likely by decreasing involucrin, claudin-1, and caspase-14 expression [65,66]. Additionally, they reduce the *Staphylococcus aureus* population and restore the balance between *Staphylococcus aureus* and *Staphylococcus epidermidis*, as involucrin is crucial for *Staphylococcus aureus* adhesion to skin cells via the staphylococcal adhesion receptor [67]. “Emollient plus” refers to emollients that contain additional active agents designed to enhance their therapeutic efficacy. Bioactive compounds such as flavonoids, riboflavins, quinones, tannins, catechins, and phenols commonly derived from botanical extracts such as *Aloe vera*, *Curcuma longa*, *Calendula officinalis*, *Matricaria chamomilla*, among others, are incorporated for their bacteriostatic and antioxidant properties [68,69]. These compounds act through mechanisms such as inactivating microbial adhesins and cell envelope transport proteins by binding to nucleophilic amino acids in these proteins, as demonstrated in vitro and in animal models [69,70,71]. However, efficacy data from only a limited number of randomized controlled trials are available for these formulations in the context of RISI; therefore, they are not currently recommended in clinical practice [72]. It is important to highlight that while plant-derived compounds are generally safe, there is a growing number of cosmetics and topical products containing whole natural botanical extracts. In susceptible individuals, these extracts might cause allergic contact dermatitis [73].

Moreover, topical probiotics, such as *Vitreoscilla filiformis* biomass (VFB) or *Bifidobacterium longum*, have been studied [74,75]. VFB is widely used in emollient products and has been proven to stimulate the production of antimicrobial peptides through the Toll-like receptor 2 (TLR2)/protein kinase C, zeta pathway (PKCζ), thus modulating the activity of free radical scavenger mitochondrial superoxide dismutase (SOD) [76,77]. Prebiotics such as fructo-oligosaccharides (FOSs), galacto-oligosaccharides (GOSs), lactosucrose, glucomannan, lactulose, isomalto-oligosaccharides, sorbitol, xylito-oligosaccharides, and xylitol are frequently incorporated into emollient formulations [74]. Limited knowledge exists regarding the efficacy of topically applied prebiotics, as they are always studied in products with complex formulations. However, they are believed to stimulate the activity of beneficial skin microbiota, thereby suppressing the expansion of pathogenic skin flora such as *Staphylococcus aureus*, among others.

The skin affected by RISI is highly susceptible to UV radiation due to disruptions in the hydrolipid barrier and alterations in the natural skin microbiota [78]. *Staphylococcus epidermidis*, for instance, produces 6-N hydroxyaminopurine (6-HAP), which inhibits UV-induced cell proliferation. *Cyanobacteria* produce mycosporine-like amino acids (MAAs) that absorb UV radiation, while *Micrococcus luteus* synthesizes an endonuclease that enhances the efficacy of DNA repair enzymes, thereby bolstering the skin’s defense against UV-induced damage. In vitro studies have shown that *Lactobacillus* species prevent the development of skin cancers due to the activity of cell wall-embedded lipoteichoic acid (LTA). Moreover, post-RT patients exhibit an elevated risk of developing both melanoma and non-melanoma skin cancers (NMSCs). Daily application of sun protection factor (SPF)-containing products is essential for all individuals; however, it is particularly significant for patients receiving RT, as the disrupted hydrolipid barrier and cutaneous microbiota increase sensitivity to UV radiation, necessitating rigorous photoprotection to mitigate potential skin damage [79,80].

### 5.2. Treatment Options and the Skin Microbiome

The management of RISI remains without universally accepted treatment protocols. Despite the extensive literature describing treatment modalities, significant disparities exist in clinical practice. The data available for acute RISI (aRISI) are considerably more substantial than those for cRISI, with minimal evidence addressing the appropriate management of cRISI [72,81].

Topical glucocorticoids (GCSs) remain the mainstay in the treatment of RISI. They have anti-inflammatory, antiproliferative, and immunosuppressive effects [82]. They suppress multiple immune cells, including neutrophils, monocytes, lymphocytes, and skin-resident Langerhans cells, through the inhibition of various pro-inflammatory cytokines such as IL-1α, IL-1β, IL-2, TNF-α, and granulocyte–macrophage colony-stimulating factor (GM-CSF) [82]. On the other hand, topical GCSs disrupt the synthesis of cholesterol, ceramides, and free fatty acids, leading to the impairment of the hydrolipid barrier [83]. This results in increased TEWL and compromises the antimicrobial function of the skin barrier. While topical GCS therapy decreases inflammation and the clinical signs of RISI, it can further impair the already damaged skin barrier due to RT. As previously noted, the microbiome in RISI is significantly less diverse, with a predominance of certain opportunistic pathogens. However, even in the absence of clinical signs of skin infection, topical GCSs reduce inflammation and promote healing [84]. Another study indicates that topical GCSs alone and the addition of topical mupirocin to topical GCSs can reduce *Staphylococcus aureus* colonization, resulting in a significant clinical improvement in patients with AD [85].

The alternative to topical GCSs could be topical calcineurin inhibitors (CIs), although it is important to note that these have not yet been extensively studied in RISI and are not included in current consensus statements and recommendations. They appear to be safe in the RT setting and, together with topical GCSs, form a cornerstone of AD treatment [86,87,88,89,90]. Experimental studies using rat models of radiotherapy-induced cystitis demonstrated that intravesical administration of tacrolimus exhibited protective effects against this condition [89]. Furthermore, patients receiving systemic administration of calcineurin inhibitors, such as those undergoing organ transplantation, did not appear to exhibit increased levels of radiotherapy-related toxicities [90]. Topical CI inhibits the activation of T cells, thereby suppressing the production of IL-2, IL-4, IL-10, interferon (IFN)-γ, and TNF-α, with no effect on Th cells and Langerhans cells [91,92]. Furthermore, topical pimecrolimus has been observed to reduce involucrin levels, thereby restoring the hydrolipid barrier and reducing the adhesion of *Staphylococcus aureus* [67].

Silver sulfadiazine and silver-containing dressings are frequently utilized in patients with aRISI and clinical signs of infection [93,94]. It is noteworthy that silver sulfadiazine should not be used for longer than 14 days, as it may slow down re-epithelization [95]. Silver exerts its antimicrobial activity by binding to bacterial DNA, thereby inhibiting the replication process [96]. Additionally, silver inhibits the microbial electron transport system and respiration. It has demonstrated efficacy against pathogenic species of bacteria commonly implicated in skin infections, such as *Staphylococcus aureus* and *Pseudomonas aeruginosa*, which are also prevalent among RISI patients [97]. As anticipated, this may also result in bacteriostatic effects on the positive, commensal bacteria on the skin. While comprehensive studies on antimicrobial silver-containing agents are lacking, research has explored the impact of silver-thread-enriched clothing on human skin [98]. Findings indicate that individuals wearing silver-containing clothing exhibit increased bacterial biomass, contradicting expectations, given silver’s antimicrobial properties. Predominant species identified include *Staphylococcus*, *Corynebacterium*, and *Cutibacterium*, associated with heightened production of monounsaturated fatty acids (MUFAs) such as myristoleic acid, contributing to elevated sebum production and skin inflammation [99]. This investigation suggests that the application of silver-containing agents in RT patients could perturb the natural microbiota of the skin, thereby compromising the integrity of the skin barrier and promoting the proliferation of pathogenic species, leading to RISI exacerbation. Table 2 summarizes the main treatment options in RISI as well as their effect on the skin microbiome.

Current recommendations suggest that there is no need to use topical or systemic antibiotics in the absence of clinical signs of infection. However, a recent study by Kost et al. indicated a significant reduction in the risk of RISI following bacterial decolonization of the nose and skin [100]. The researchers used chlorhexidine, which is known to be allergenic and to damage the skin barrier. Therefore, we propose using sodium hypochlorite baths, which are successfully used in patients with atopic dermatitis and recurrent bacterial skin infections and are currently considered the least aggressive antiseptic [63,101,102]. Hypochlorous acid non-selectively eradicates *Staphylococcus aureus* along with other bacteria such as *Staphylococcus pyogenes*, *Pseudomonas aeruginosa*, *Propionibacterium acnes*; fungi, such as *Candida* species; and viruses [102,103,104]. Additionally, it exhibits anti-inflammatory properties by reducing the levels of IL-1, IL-4, IL-6, IL-12, and IL-13, as well as TNF-α. Importantly, it does not significantly affect the TEWL parameter but improves the stratum corneum integrity, thus reinforcing the skin barrier [102,105]. Furthermore, it alleviates itching by decreasing the levels of pruritogenic cytokines and inhibiting mast cell degranulation [106,107].

## 6. Conclusions and Future Perspectives

Human skin is home to a vast number of different species of bacteria, viruses, and fungi. Its complex microbiome is crucial for proper barrier function, and dysbiosis has been associated with the pathogenesis of numerous skin disorders and diseases. RISI has recently emerged as being characterized by significant alterations in the abundance of certain bacterial species. Given the complex symbiotic and pathomechanistic relationships of the development of RISI, which includes a cascade of immunological processes and damage to the epidermal barrier, it is crucial to further explore the mutual relationships between skin microorganisms before, during, and after RT to provide valuable insights into the dynamics of microbial communities in response to radiation exposure. Importantly, it remains unknown whether microbial cells or their metabolites impact skin cells and surrounding cells like immune, neuronal, and other sensory cells as well as sweat and other activities. Current evidence primarily focuses on taxonomic shifts in healthy and irradiated skin. However, there is an ongoing need for metabolomic or transcriptomic studies, which are essential to link microbiota changes to functional pathways such as SCFA production, molecular pathway alterations, gene expression, and cytokine modulation. These functional insights are crucial to determine how the altered microbiota influences host physiology beyond compositional changes. Currently, it remains difficult to determine which observed microbial shifts are causative, compensatory, or correlative in the context of RISI. Moreover, although some data come from ex vivo or in vivo studies on humans, these are still limited. Most studies have been conducted on animal models, which provide insights into potential pathways but do not fully elucidate the complexities. Therefore, we suggest conducting more studies on humans or using human-sampled tissues. Further research should also explore the long-term effects of irradiation on the destabilization of skin microbiota. In addition, the development of microbiome-based interventions with either probiotics or bacterial metabolites should be a future therapeutic target to prevent and manage RISI.

## Figures and Tables

**Figure 1 ijms-26-05022-f001:**
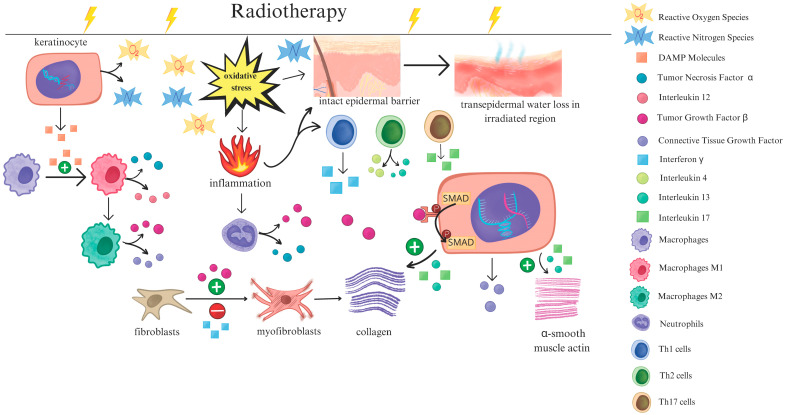
A schematic representation of the subsequent changes in immune cells, skin cells, and skin barrier following radiotherapy.

**Table 1 ijms-26-05022-t001:** A summary of the findings regarding microbiota in radiation-induced skin injury (RISI).

Authors and Year of Publication	Research Group	Time of Sample Collection	Results	Reference
Ramadan et al., 2021	78 cancer patients and 20 control subjects with no RT history	RISI recovery after 2, 3, 4, 5, 6, or 7 weeks, or chronic ulcers	RISI group—predominance of Firmicutes and Proteobacteria, predominance of *Klebsiella*, *Staphylococcus*, or *Pseudomonas* Group with a longer healing of RISI—predominance of *Pseudomonas*, *Staphylococcus*, and *Stenotrophomonas*. Chronic ulcers—predominance of Proteobacteria, low abundance of Firmicutes, predominance of *Klebsiella* or *Pseudomonas*, *Cutibacterium* and *Stenotrophomonas,* coexistence of *Pseudomonas*, *Staphylococcus*, and *Stenotrophomonas*	[10]
Huang et al., 2022	29 male rats	healthy subjects (control group)—skin samples taken before RT aRISI model—2 weeks after RT	predominance of Firmicutes in aRISI (*Streptococcus*, *Staphylococcus*, *Acetivibrio ethanolgignens*, *Peptostreptococcus*, and *Anaerofilum*)	[56]
	patient data from BioProject 665254	Study group—after RT Control group of healthy subjects with no RT history	Greater abundance of *Klebsiella*, *Pseudomonas*, and *Staphylococcus* after RT compared with control group. Chronic ulcers were associated with predominance of Proteobacteria and a low abundance of Firmicutes after RT.	
Kost et al., 2023	76 patients with head and neck or breast cancer	before and after RT	Among 16 patients with positive nasal *Staphylococcus aureus* colonization prior to RT, 10 of them developed grade 2 or higher aRISI (34.5% of all patients with grade 2 or higher and 62.5% of patients with positive colonization), and 6 of them developed grade 1 (12.8% of all patients with grade 1 and 37.5% of all patients with positive colonization). Among 60 patients with negative nasal *Staphylococcus aureus* colonization prior to RT, 19 of them developed grade 2 or higher aRISI (65.5% of all patients with grade 2 or higher but 31.6% of patients with negative colonization), 41 of them developed grade 1 (87.2% of all patients with grade 1 and 68.3% of all patients with negative colonization).	[60]
Shi et al., 2023	100 patients with breast cancer	before and after RT	Significantly higher abundance of *Ralstonia*, *Truepera*, and *Methyloversatilis* genera and lower abundance of *Staphylococcus* and *Corynebacterium* genera in patients with no/mild aRISI (RTOG 0/1) compared with patients with severe aRISI (RTOG 2 or higher) both before and after RT.	[57]
Hülpüsch et al., 2024	20 patients with breast cancer	before and after RT	Low (<5%) abundance of commensal bacteria *Staphylococcus epidermidis*, *Staphylococcus hominis*, *Cutibacterium acnes* before RT was associated with the development of severe aRISI with an accuracy of 100%. Overgrowth of skin bacteria before the onset of severe aRISI during or after RT.	[58]
Miyamae et al., 2025	9 head and neck cancer patients who received chemoradiotherapy	before RT	Lower abundance of *Staphylococcus hominis* and *Staphylococcus aureus* before RT in severe aRISI compared with the non-severe group.	[59]

RISI—radiation-induced skin injury, aRISI—acute radiation-induced skin injury, RT—radiotherapy, RTOG—Radiation Therapy Oncology Group.

**Table 2 ijms-26-05022-t002:** Summary of the main radiation-induced skin injury (RISI) treatment approaches and their effect on the skin microbiome.

Treatment Option	Effect on the Skins’ Microbiome	Reference
Emollients	Reduction in pathogenic *Staphylococcus aureus* colonization with simultaneous increase in commensal (*Staphylococcus epidermidis*) skin microbiota	[65,66]
Topical GCSs	Reduction in pathogenic *Staphylococcus aureus* colonization	[82,85]
Topical CIs	Reduction in pathogenic *Staphylococcus aureus* colonization	[67,91]
Silver-containing agents	Reduction in both pathogenic (*Staphylococcus aureus*, *Pseudomonas aeruginosa*) and commensal skin microbiota	[97,98]

GCSs—glucocorticoids, CIs—calcineurin inhibitors.

## Data Availability

All data are available upon request.

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
