# Peer review of "Skin Microbiome and Radiation-Induced Skin Injury: Unraveling the Relationship, Mechanisms, and Therapeutic Implications"

_ijms, 2025, doi:10.3390/ijms26115022_

Round 1
Reviewer 1 Report
Comments and Suggestions for Authors
This is a well-written and interesting manuscript, aiming to present the current findings on the interplay between the skin microbiome and radiation-induced skin damage, as well as to discuss potential therapeutic strategies for preventing this condition. The tables presented by the authors are illustrative and clear. I would only suggest that the authors speculate further on the potential cellular and molecular mechanisms underlying radiation-induced skin damage in a separate paragraph to make the message clearer for readers.
Author Response
Dear Reviewer,
Thank you very much for reviewing our manuscript. We sincerely appreciate your time and effort. Please find our response to your comments below written in red.
This is a well-written and interesting manuscript, aiming to present the current findings on the interplay between the skin microbiome and radiation-induced skin damage, as well as to discuss potential therapeutic strategies for preventing this condition. The tables presented by the authors are illustrative and clear. I would only suggest that the authors speculate further on the potential cellular and molecular mechanisms underlying radiation-induced skin damage in a separate paragraph to make the message clearer for readers.
We have added more detailed information of the cellular and molecular pathways involved in radiation-induced skin damage in a dedicated paragraph.
Thank you for the thorough analysis of our manuscript and for providing us with constructive comments. We have addressed your every suggestion, and we hope that the implemented modifications improve the quality of our article and reach the point of acceptance for publication.
Yours faithfully,
Authors
Reviewer 2 Report
Comments and Suggestions for Authors
This work is really interesting because it digs into how the skin microbiome and radiation-induced skin injury (RISI) affect each other. It tries to figure out how changes in the skin’s microbial community can make RISI worse or better. Plus, it looks at ways we might be able to tweak the skin microbiome to help protect against RISI.
What makes this review special is that it takes a deep dive into a topic that hasn’t been explored much before. By pulling together the latest research, the authors shed light on how radiation therapy (RT) messes with the skin’s microbial community and what that means for patients. The part about potential treatments that focus on the skin microbiome is especially cool and relevant.
Compared to other stuff out there, this manuscript gives a fresh look at the complicated relationship between RT, the skin microbiota, and RISI. It brings together findings from different studies and offers new ideas about how the makeup of the skin microbiome before RT might predict how bad RISI will be, and why it’s so important to keep the skin microbiome balanced during and after RT. This adds a lot to our understanding of how RISI works and points the way to new treatment ideas.
In conclusion, this manuscript offers valuable insights into the emerging field of skin microbiome and RISI, highlighting the need for further research to develop effective therapies targeting the skin microbiome. Major adjustments in methodology suggestions and reference updates could further enhance its contribution to the field.
- The experimental procedures lacked standardization. The cited studies differed in sampling sites (skin on the irradiated side vs. contralateral skin), timing (before/after irradiation), and microbiological analysis methods (16S rRNA vs. shotgun sequencing). These deficiencies seriously affected the credibility of the experimental results.
- Many studies lack controls for confounders like antibiotics, chemotherapy, or topical treatments. Including longitudinal data from non-RT cancer patients would clarify RT-specific effects.
- Current evidence focuses on taxonomic shifts; metabolomic or transcriptomic analyses are needed to link microbial changes to functional pathways (e.g., SCFA production, cytokine modulation).
- Key questions about microbiome-driven immune modulation are addressed through cited animal models (e.g., S. epidermidis enhancing wound healing via T-cell activation), though human validation is limited.
- Smaller studies (e.g., n = 9 in Miyamae et al.) limit generalizability. The inclusion of both animal models and human data balances this but warrants cautious interpretation.
- Table 1: Clearly summarizes study designs and findings but omits methodological details (e.g., sequencing depth, statistical corrections). Adding sample sizes and key confounders (e.g., concurrent therapies) would improve reproducibility assessment.
- The references are generally appropriate, covering basic research and recent clinical data. However, they seem to be not comprehensive enough, and the following key literature is omitted. Please read and supplement the citations.
A.Hu, D., et al. (2024). "Phaeohyphomycosis caused by Corynespora cassiicola, a plant pathogen worldwide." Mycology 15(1): 91-100.
- Kaki, R. (2023). "Risk factors and mortality of the newly emerging Candida auris in a university hospital in Saudi Arabia." Mycology 14(3): 256-263.
- Zhong, J., et al. (2024). "Role of Dectin-1 in immune response of macrophages induced by Fonsecaea monophora wild strain and melanin-deficient mutant strain." Mycology 15(1): 45-56.
- The immune-microbiome interaction section should be strengthened by including works on RT’s impact on cutaneous immune cells and their crosstalk with commensals. Additionally, seminal papers on microbiome metabolites are underrepresented.
Author Response
Dear Reviewer,
Thank you very much for reviewing our manuscript. We sincerely appreciate your time and effort. Please find our response to your comments below written in red.
This work is really interesting because it digs into how the skin microbiome and radiation-induced skin injury (RISI) affect each other. It tries to figure out how changes in the skin’s microbial community can make RISI worse or better. Plus, it looks at ways we might be able to tweak the skin microbiome to help protect against RISI.
What makes this review special is that it takes a deep dive into a topic that hasn’t been explored much before. By pulling together the latest research, the authors shed light on how radiation therapy (RT) messes with the skin’s microbial community and what that means for patients. The part about potential treatments that focus on the skin microbiome is especially cool and relevant.
Compared to other stuff out there, this manuscript gives a fresh look at the complicated relationship between RT, the skin microbiota, and RISI. It brings together findings from different studies and offers new ideas about how the makeup of the skin microbiome before RT might predict how bad RISI will be, and why it’s so important to keep the skin microbiome balanced during and after RT. This adds a lot to our understanding of how RISI works and points the way to new treatment ideas.
In conclusion, this manuscript offers valuable insights into the emerging field of skin microbiome and RISI, highlighting the need for further research to develop effective therapies targeting the skin microbiome. Major adjustments in methodology suggestions and reference updates could further enhance its contribution to the field.
- The experimental procedures lacked standardization. The cited studies differed in sampling sites (skin on the irradiated side vs. contralateral skin), timing (before/after irradiation), and microbiological analysis methods (16S rRNA vs. shotgun sequencing). These deficiencies seriously affected the credibility of the experimental results.
We have included these information and conclusions in paragraph 4.
2. Many studies lack controls for confounders like antibiotics, chemotherapy, or topical treatments. Including longitudinal data from non-RT cancer patients would clarify RT-specific effects.
Thank you for your suggestion. However, the focus of this review is to elucidate radiation-induced skin injury specifically. Incorporating information regarding non-radiotherapy cancer patients would deviate from our core objective, potentially compromising the integrity of our work and introducing confusion to readers.
3. Current evidence focuses on taxonomic shifts; metabolomic or transcriptomic analyses are needed to link microbial changes to functional pathways (e.g., SCFA production, cytokine modulation).
We have included these conclusions in paragraph 6.
4. Key questions about microbiome-driven immune modulation are addressed through cited animal models (e.g., S. epidermidis enhancing wound healing via T-cell activation), though human validation is limited.
Thank you for your suggestion. We have expanded on this topic.
5. Smaller studies (e.g., n = 9 in Miyamae et al.) limit generalizability. The inclusion of both animal models and human data balances this but warrants cautious interpretation.
We have included these information and conclusions in paragraph 4.
6. Table 1: Clearly summarizes study designs and findings but omits methodological details (e.g., sequencing depth, statistical corrections). Adding sample sizes and key confounders (e.g., concurrent therapies) would improve reproducibility assessment.
The information about sample sizes is provided. Where applicable, we mention chemoradiotherapy. As the cited studies focused on microbiota alterations following radiotherapy, the information about concurrent therepies, such as surgery or specific chemotherapy is omitted.
7. The references are generally appropriate, covering basic research and recent clinical data. However, they seem to be not comprehensive enough, and the following key literature is omitted. Please read and supplement the citations.
- Hu, D., et al. (2024). "Phaeohyphomycosis caused by Corynespora cassiicola, a plant pathogen worldwide." Mycology 15(1): 91-100.
- Kaki, R. (2023). "Risk factors and mortality of the newly emerging Candida auris in a university hospital in Saudi Arabia." Mycology 14(3): 256-263.
- Zhong, J., et al. (2024). "Role of Dectin-1 in immune response of macrophages induced by Fonsecaea monophora wild strain and melanin-deficient mutant strain." Mycology 15(1): 45-56.
Thank you for suggesting the significant manuscripts. We have expanded on the information regarding fungal skin colonization in paragraph 3, citing the aforementioned literature.
- The immune-microbiome interaction section should be strengthened by including works on RT’s impact on cutaneous immune cells and their crosstalk with commensals. Additionally, seminal papers on microbiome metabolites are underrepresented.
We have added more detailed information of the cellular and molecular pathways involved in radiation-induced skin damage in a dedicated paragraph. We have expanded on microbiome metabolites in paragraph 3.
Thank you for the thorough analysis of our manuscript and for providing us with constructive comments. We have addressed your every suggestion, and we hope that the implemented modifications improve the quality of our article and reach the point of acceptance for publication.
Yours faithfully,
Authors
Round 2
Reviewer 2 Report
Comments and Suggestions for Authors
The authors have undertaken meticulous and earnest revisions to the manuscript,resulting in a noticeable enhancement in its quality.I recommend that the manuscript be accepted for publication.
Author Response
Dear Reviewer,
Thank you very much for reviewing and accepting our manuscript. We appreciate your time and effort taken in improving the quality of our work.
Yours faithfully,
Authors